# Mechanical and Structural Adaptation of the Pulmonary Root after Ross Operation in a Murine Model

**DOI:** 10.3390/jcm11133742

**Published:** 2022-06-28

**Authors:** Claudia Cattapan, Mila Della Barbera, Arben Dedja, Piero Pavan, Giovanni Di Salvo, Jolanda Sabatino, Martina Avesani, Massimo Padalino, Alvise Guariento, Cristina Basso, Vladimiro Vida

**Affiliations:** 1Pediatric and Congenital Cardiac Surgery Unit, Department of Cardiac, Thoracic and Vascular Sciences, University of Padua, 35121 Padua, Italy; claudia.cattapan@gmail.com (C.C.); arben.dedja@unipd.it (A.D.); massimo.padalino@unipd.it (M.P.); alvise.guariento@hotmail.com (A.G.); 2Cardiovascular Pathology Unit, Department of Cardiac, Thoracic and Vascular Sciences, University of Padua, 35121 Padua, Italy; mila.dellabarbera@unipd.it (M.D.B.); cristina.basso@unipd.it (C.B.); 3Department of Industrial Engineering, University of Padua, 35131 Padua, Italy; piero.pavan@unipd.it; 4Pediatric Cardiology Unit, Department of Children and Woman’s Health, University of Padua, 35121 Padua, Italy; giovanni.disalvo@unipd.it (G.D.S.); jolesbt@hotmail.it (J.S.); martina.avesani@studenti.unipd.it (M.A.)

**Keywords:** Ross procedure, aortic valve disease, autograft, congenital heart disease

## Abstract

Background: The major limitation to the Ross operation is a progressive autograft dilation, possibly leading to reoperations. A murine model was created to evaluate pulmonary artery graft (PAG) adaptation to pressure overload. Methods: Lewis rats (n = 17) underwent heterotopic surgical implantation of a PAG, harvested from syngeneic animals (n = 17). A group of sham animals (n = 7) was used as a control. Seriated ultrasound studies of the PAG were performed. Animals were sacrificed at 1 week (n = 5) or 2 months (n = 15) and the PAG underwent mechanical and histopathological analyses. Results: Echography showed an initial increase in diameter (*p* < 0.001) and a decrease in peak systolic velocity (PSV). Subsequently, despite no change in diameter, an increase in PSV was observed (*p* < 0.01). After 1 week, the stiffness of the PAG and the aorta were similar, while at 2 months, the PAG appeared more rigid (*p* < 0.05). PAG’s histological analysis at 2 months revealed intimal hyperplasia development. The tunica media showed focal thinning of the elastic lamellae and normally distributed smooth muscle cells. Conclusions: We demonstrated a stiffening of the PAG wall after its implantation in systemic position; the development of intimal hyperplasia and the thinning of the elastic lamellae could be the possible underlying mechanism.

## 1. Introduction

Aortic valve replacement is a common procedure in the adult population. However, the options available for pediatric patients are usually limited to mechanical valves. This is due to the early calcification of biological valves in this population [1,2]. Nevertheless, mechanical valves are still a suboptimal choice due to the risk associated with life-long anticoagulation therapy and the lack of growth potential, which predisposes patients to further reoperations [3,4].

The Ross operation represents an alternative to prosthetic heart valves and contemplate the transfer of the pulmonary valve and root to the aortic position (with coronary’s ostia reimplantation) and the replacement of the first one usually with a homograft. This procedure has regained new interest due to its exceptional long-term results, being one of the most attractive choices for the pediatric population [5]. Despite this, its use remains limited due to the possible dilation of the pulmonary autograft and subsequent aortic regurgitation [6]. At present, there is little understanding of the changes that occur on the wall of the autograft when this is placed in a systemic position. For this reason, animal models represent an interesting platform to study the mechanism behind this process.

In this study, we established a murine model of the Ross operation by pulmonary artery graft (PAG) implantation in a systemic position with the aim of investigating the mechanical and structural adaptation of the pulmonary root wall under increased pressure load.

## 2. Materials and Methods

### 2.1. Animal Care and Bio Safety

This investigation was conducted in accordance with the National Institutes of Health’s Guide for the Care and Use of Laboratory Animals and was approved by the Animal Care and Use Committee of the University of Padua (Protocol 700/2018-PR, issued on 13 September 2018, DL n. 16/92 art. 5). All animals received humane care in compliance with the Guide for the Care and Use of Laboratory Animals.

### 2.2. Experimental Model

A total of 41 syngeneic Lewis rats (LEW/Han^®^Hsd) were purchased at the age of 10 weeks (Harlan Laboratories, San Pietro al Natisone, Udine, Italy). Animals were randomly divided into three experimental groups (Figure 1): a transplant group (TG) underwent heterotopic PAG implantation at the level of the infrarenal abdominal aorta (n = 17); a donor group (DG) underwent pulmonary root harvesting for the TG group (n = 17); a sham-operated group (SOG) was used as a control group (n = 7).

#### 2.2.1. Anesthesia Protocol

All operations were performed under clean conditions. Both male and female adult Lewis rats were used. The same anesthesia protocol was used for all the groups.

An intraperitoneal injection of 5 mg/kg tramadol (CONTRAMAL^®^, Formenti Srl, Milan, Italy) was performed 15 min before the surgery. A single dose of intramuscular Gentamicin (5 mg/kg) was administered immediately before surgery.

The rats were first placed in a Plexiglas chamber where 4% sevoflurane (SEVORANE^®^, Abbott SpA, Campoverde, Italy) in 1 L/min of oxygen was supplied to perform the anesthesia induction.

Once sedated, the animal was placed on the surgical table and connected to a Fluovac Sevoflurane/Halothane Scavenger unit (Harvard Apparatus Ltd., Kent, UK) with an absorber filter where the anesthesia maintenance was performed using sevoflurane at 2.0–2.5% in 1 L/min of oxygen throughout the procedure.

Finally, the level of anesthesia was evaluated before performing the procedure to assess absence of response to a noxious stimulus.

#### 2.2.2. Pulmonary Graft Harvesting

After the sedation and the correct positioning of the sedated donor animal on the surgical table, the chest was shaved, and the pulmonary artery diameter was measured using an ultrasound scan. A xipho-pubic incision was then performed, and a heparin bolus was administered in inferior vena cava. A bilateral thoracotomy was carried out to expose the mediastinal structures. A pericardiectomy and a thymectomy were performed to isolate the aorta, which was resected distally to the innominate artery.

In this group of animals, the euthanasia was performed by cooling, and the heart was cooled down using a saline solution at 4 °C until it stopped beating with a visual assessment.

Finally, the pulmonary artery was resected the closest possible to its two branches, distally, including a portion of right ventricle proximally.

The explanted block was then transferred to the table and a resection of the right ventricle was performed, leaving only 1 mm of muscle left. Finally, the PAG was conserved in a saline solution and placed in refrigerated conditions at 5 °C.

#### 2.2.3. Recipient Operation

The animal was sedated using sevoflurane (at 3% for induction and at 1.5–2% for maintenance) and an adequate analgesic therapy was provided (subcutaneous tramadol 5 mg/kg); it was then placed on the operating table on spontaneous breathing. A xipho-pubic incision was performed and two mini-retractors were used to expose the abdominal cavity.

The infra-renal abdominal aorta was isolated, and any collateral branches were ligated using a 6/0 silk suture. A 2/0 silk suture was used to place a gentle traction on the aorta to isolate it from the perivasal adipose tissue and the inferior vena cava.

Two Yasargil clips were then placed between the two renal arteries and the iliac bifurcation, leaving 1.5 cm of distance between them. The vessel was cut in the middle of the two clips and an irrigation of saline solution with heparin 1 UI/mL was performed to prevent any thrombotic events.

The graft was anastomosed termino-terminally using a 10/0 Prolene suture placing the ventricular end proximally. First, two landmark sutures were placed diametrically to the vase circumference at both sides of the graft. Then, the proximal suture was performed starting with the posterior wall and using a continuous sewing. Finally, the distal end of the graft was anastomosed with the abdominal aorta using the same technique.

Once the sutures were performed, the distal clip was released first to let the graft be perfused with a retrograde low-pressure blood in order to check the anastomosis. An accurate hemostasis was performed using hemostatic sponges at the end of the procedure to avoid any possible site of bleeding.

#### 2.2.4. Sham-Operated Group Intervention

The sham-operated group only underwent the surgical stress. The same pre-operatory and anesthesiologic procedures were performed, and again, a xipho-pubic incision was used to expose the abdominal aorta. The latter was resected in the middle between the two renal arteries and the iliac bifurcation, and then the two ends were directly re-anastomosed.

#### 2.2.5. Post-Operatory Management

Immediately after the operation, the animals were placed under a heating lamp and monitored until complete awakening. Pain was controlled with tramadol (5 mg/kg bis in die during the first 48 h, then as needed), and daily psychophysics assessment was performed by expert veterinarians using the pain score system based on clinical manifestations of distress.

At the end of the in vivo phase, animals of the TG and the SOG were euthanized using CO_2_ for a few minutes. In the TG’s animals, the pulmonary graft and a small portion of the near aorta were explanted and divided in two halves; as for the SOG, the same procedure was performed including the aorta just before and after the suture line. The proximal half was conserved in 4% formalin in 0.1 M phosphate buffer and used for the histopathological studies; the distal one was placed in a physiological solution at 5 °C and underwent mechanical traction studies.

### 2.3. Ultrasound Studies

A high-resolution ultrasound machine (Vevo^®^ 2100 Imaging System, VisualSonics, Toronto, ON, Canada) with a 13–24 MHz probe was used for 1 week, 1 month, and 2 month ultrasound scans in the TG and 1 week and 2 month scans in the SOG. Animals were anesthetized with 3% sevoflurane, and temperature-controlled anesthesia was maintained with 1.5% sevoflurane. Prior to the scans, animals were placed supine on an adjustable stage, and their hair on the ventral abdomen was removed with a depilatory cream. Next, transmission ultrasound gel was applied to the exposed skin surface. Transaxial and longitudinal ultrasound data were acquired by placing the transducer perpendicular to the animal and in contact with the gel during the entire examination. The angle was adjusted as needed to allow for optimal visualization of the aorta in the long and short axes. Anatomical landmarks, such as the inferior vena cava, aortic bifurcation, and renal veins, were used to allow for anatomical orientation, while the presence or absence of vessel wall movement was used to discriminate arterial flow from a venous flow. Two-dimensional high-resolution cine loops in brightness mode (mode B), motion mode (mode M), pulsed wave (PW), and color Doppler were recorded in the abdominal aorta. The transducer angle and PW Doppler angle (30–60°) were adjusted to correctly assess the amplitude and direction of blood flow in the selected area. The diameter of the native aorta was measured upstream and downstream of the PAG. The PAG diameter was measured by cine B-mode loops. Doppler assessment of abdominal aortic blood flow was performed from an abdominal longitudinal view and included peak systolic velocity and end-diastolic velocity. In SOG animals, the assessment of the abdominal aorta (diameter and Doppler flow) was performed halfway between the origin of the renal arteries and the iliac bifurcation. A single operator performed all ultrasound and Doppler evaluations.

### 2.4. Euthanasia

Animals were euthanized in a CO_2_ chamber in accordance with the American Physiological Society’s Guiding Principles for the Care and Use of Vertebrate Animals in Research and Training protocol. In the TG, 5 animals were sacrificed 1 week after the procedure, while 10 animals were euthanized 2 months after PAG implantation. After euthanasia, all cardiac great vessels and the infrarenal abdominal aorta were harvested for ex vivo mechanical tests and histological analyses.

### 2.5. Ex Vivo Mechanical Tests

An advanced experimental setup (Planar Biaxial Test Bench Test Instrument, Bose^®^ Electro-Force^®^, New Castle, DE, USA) was used to test the mechanical properties of the vessels. A quasi-static uniaxial tensile test in the circumferential direction was performed to evaluate the structural modification of major interest [7].

These studies were performed, in the TG, 1 week and 2 months after the operation to compare short- and long-term changes, using the nearby aorta and native pulmonary artery as controls. The distal half of the PAG was first cut lengthwise and then perpendicularly to obtain rectangular tissue samples. The length of the samples used for the mechanical tests was 2.5–5 mm and the width was 2–3 mm. The samples were then fixed to the grips of the machine by interposing balsa wood fixed with a cyanoacrylate glue at the ends of the sample and equally distributing the force exerted by the grips on the fabric.

The configuration included high-performance electromagnetic induction linear actuators, which allow evaluation of sample elongations with an accuracy of 1 μm. A load cell with a force resolution of 0.001 N was used. All tests were conducted at room temperature (25 °C), ensuring hydration of the tissue samples with a continuous pipetting of saline. The uniaxial tensile tests were performed under displacement control, with an elongation rate corresponding to a nominal strain rate of 0.01 s^−1^, where the nominal strain is defined as the ratio of the sample elongation to its initial length.

The results are expressed in graphs representing the membrane force with respect to the nominal strain. The membrane force represents the ratio between the tensile force (F) and the initial width of the sample (w0). From these data, we deduced the membrane stiffness of the samples, defined as the ratio of the membrane force to the nominal strain at a specified value of nominal strain. Therefore, the membrane stiffness represents the secant of the membrane force with respect to the nominal strain curve at the specified value of the nominal strain.

This is a reliable indicator of the mechanical behavior of the wall, combining the mechanical properties of the fabric and the thickness of the wall [8]. The tensile test data were considered up to a maximum nominal strain value (assumed as 0.4). This interval was chosen on the basis of preliminary studies performed by our group which have shown that the native vascular tissue can possibly suffer damage beyond this value.

### 2.6. Gross and Histopathological Analysis

After macroscopic examination, the PAG and SOG’s aorta samples were photographed with an Olympus SZX12 stereomicroscope (Olympus Europa, Hamburg, Germany). Next, the XPert80 preclinical X-ray imaging system (Kubtec Medical Imaging, Stratford, CT, USA) was used to identify any calcium deposits. A semiquantitative score was used in this contest as follows: absent calcification (=0); focal, point-like, with a diameter of less than 1 mm (=1); focal, with diameter greater than 1 mm or multi-point deposits (=2); multiple deposits with a diameter greater than 1 mm (=3); and massive deposit of calcium (=4) [9].

The samples were then cut lengthwise, dehydrated in increasing concentrations of ethanol solutions, and embedded in paraffin. Sections with a thickness of 3-4 μm were obtained using a Microm HM 450 microtome (Thermofisher Scientific Inc., Waltham, MA, USA) and then stained with hematoxylin-eosin, Weigert-Van Gieson, Heidenheim modified Azan-Mallory, Alcian-PAS, and picrosirius red. A qualitative assessment of necrosis on hematoxylin-eosin-stained samples was performed as follows: presence (=1); absence (=0).

#### 2.6.1. Immunohistochemistry

Other paraffin sections were subjected to antigen retrieval by heating in a microwave oven in 0.01 M citrate buffer (pH 6.0) for 10 min, quenching of endogenous peroxidase in 0.3% hydrogen peroxide in buffer phosphate for 20 min, and incubation with the primary antibodies as reported in Appendix A. Primary antibody incubation was followed by Envision HRP Kit Complex (Dakocytomation, Glostrup, Germany). The sections were then counterstained with Meyer’s hematoxylin.

The percentage of inflammatory cells, macrophages (CD45 and CD68 markers), smooth muscle cells (alpha smooth muscle actin, α-SMA), and the presence of endothelial (von Willebrand factor, vWF) cells and activated endothelial cells (inducible nitric oxide synthase, iNOS) were assessed.

#### 2.6.2. Morphometric Analysis

Histological stained slides were photographed and then examined under an optical microscope (Zeiss Axioplan 2, Carl Zeiss, Oberkochen, Germany), equipped with a digital camera (AxioVision, Carl Zeiss, Oberkochen, Germany), and software for image analysis (Image PRO-Plus 5.1; Media Cybernetics, Silver Spring, MA, USA). At least three consecutive 100x non-overlapping fields were randomly chosen for the following measurement parameters: mean tunica media thickness on Weigert-Van Gieson colored sections (mm); cell density (number of cells per mm^2^) in both tunica media and intimal hyperplasia on hematoxylin-eosin sections; elastic lamellae density (number of elastic lamellae per mm^2^) in the tunica media on Weigert-Van Gieson sections; percentage of fibrosis in both PAG and the tunica media of the latter on Heidenheim modified Azan-Mallory sections.

#### 2.6.3. Transmission Electron Microscopy

Sections of the PAG fragments were fixed with 2.5% glutaraldehyde in 0.1 M/L phosphate buffer (pH 7.2) for 4 h at +4 °C and post-fixed with tetroxide 1% osmium in the same buffer for 1 h at room temperature. The samples were then dehydrated in increasing concentrations of ethanol and embedded in epoxy resin. After the evaluation of 1 μm thick semi-thin sections, 60–70 nm thick ultra-thin sections were obtained. Other PAG fragments were processed for post-inclusion immunogold labeling as follows: after fixation with 0.5% glutaraldehyde and 2% paraformaldehyde in 0.1 M/L phosphate buffer (pH 7.2) and dehydration in increasing concentrations of ethanol, samples were embedded in LR White acrylic resin (London Resin Company, London, UK). After selecting the area of interest through semi-thin sections, ultra-thin sections were placed on nickel grids, hydrated in distilled water, and incubated with buffered saline with Tris (TBS) containing 0.02 M of glycine, pH 7.2, and normal goat serum to block non-specific sites. The sections were then incubated for 12 h at 4 °C with a human anti-α-SMA monoclonal mouse (1:20) diluted in TBS containing 0.1% BSA, pH 7.2. After washing with TBS/0.1% BSA, the sections were incubated with 10 nm colloidal gold conjugated anti-mouse IgG (Sigma Chemical Co., St. Louis, MO, USA) for 2 h at room temperature. After washing with TBS/0.1% BSA, the sections were washed in distilled water. All ultra-thin sections were contrasted with uranyl acetate and lead nitrate and examined under a Hitachi H-7800 transmission electron microscope (Hitachi, Tokyo, Japan).

### 2.7. Statistical Analysis

Normality of all continuous variables was tested using the Shapiro–Wilk test and graphically assessed by histograms and Q-Q plots. Continuous variables are expressed as mean ± standard deviation. Categorical data are presented using frequencies and percentages. Longitudinal analysis for between-group comparisons was performed using two-way repeated measures analysis of variance (ANOVA) and by fitting mixed-effects linear regression models. When a significant F-test was obtained with two-way repeated measures ANOVA, a Bonferroni-adjusted post hoc analysis was used to assess pairwise differences between groups. A *t*-test or Kruskal–Wallis test was utilized for the comparison between groups in the case of mechanical data. To reduce the probability of false-positive results (Type I error) due to the multiple comparisons, Dunn’s post hoc analysis was applied to control family-wise error to α < 0.05. All reported tests are two-tailed. Statistical analyses were performed using STATA version 15.1 (Stata Corp LLC, College Station, TX, USA).

## 3. Results

### 3.1. Study Population

A total of 22 male (53.7%) and 19 female (46.3%) rats were used. In the TG, all the animals were male (100%); in the DG, only females (100%) were employed; in the SOG, both males (5/7; 71%) and females (2/7; 29%) were used. The median weight at surgery was 376 ± 27 g in the TG, 374 ± 42 g in the SOG (*p* = 0.9), and 323 ± 33 g in the DG (*p* = 0.002).

The overall survival rate was 92% (88% in the TG and 100% in the SOG): 2 animals in the TG died 12 and 51 days after surgery, respectively. Median follow-up time was 64 ± 4 days in 10 animals in the TG and 63 ± 3 days in the SOG (*p* = 0.68). The remaining 5 animals in the TG underwent a follow-up of 7 ± 1 days. Median body weight at sacrifice was 399 ± 15 g in TG and 350 ± 28 g in SOG (*p* < 0.001).

### 3.2. Ultrasound Studies

An increase in PAG diameter was found, especially in the short term. This was 3.21 ± 0.06 mm in the native position, 3.78 ± 0.29 mm 1 week after implantation, 3.88 ± 0.25 mm after 1 month, and 4.03 ± 0.29 mm after 2 months (*p* < 0.001). In contrast, the SOG showed no signs of dilation (*p* = 0.17), with a diameter of 1.52 ± 0.15 mm at baseline, 1.59 ± 0.22 mm at 1 week, and 1.74 ± 0.17 mm at 2 months. A significant difference between the groups was found over the entire study period (*p* < 0.001) (Figure 2).

A change in peak systolic velocity (PSV) was observed, with a decrease after 1 week (from 391.7 ± 35.5 mm/s to 264.3 ± 59.6 mm/s) and a subsequent increase at 1 and 2 months (414.3 ± 83.4 mm/s and 403.5 ± 66.1 mm/s, respectively) (*p* < 0.01). In SOG rats, the PSV was 422.3 ± 38.0 mm/s at baseline (*p* = 0.24), 415.6 ± 17.9 at 1 week (*p* < 0.001), 433.85 ± 32.1 at 1 month (*p* = 0.62), and 408.7 ± 21.5 mm/s at 2 months (*p* = 0.86). The end-diastolic velocity was similar between the two groups throughout the study period (*p* = 0.13): 35.2 ± 5.6 mm/s vs. 28.3 ± 4.6 mm/s at baseline, 29.9 ± 6.6 mm/s vs. 34.7 ± 11.0 mm/s at 1 week, and 45.7 ± 11.7 mm/s vs. 48.9 ± 6.1 at 2 months.

### 3.3. Ex Vivo Mechanical Tests

A comparison of the mechanical response of the aorta and PAG at 1 week (n = 5) and 2 months (n = 10) was performed and is expressed as membrane force versus nominal strain (Figure 3). From these data, we deduced an increase in the PAG wall stiffness over the study period (*p* = 0.031).

Native pulmonary artery and aorta were used as a control for the PAG at 1 week (n = 5) and 2 months (n = 10) (Figure 4). At 1 week from the operation, there was a significant difference between the membrane stiffness of the native pulmonary artery and aorta (*p* = 0.029), while the membrane stiffness of the PAG and aorta did not differ (*p* = 0.91). However, after 2 months, an increase in this mechanical property of the PAG was seen, showing higher values than those of the aorta and the native pulmonary artery, respectively (*p* = 0.038 and *p* = 0.013, respectively).

### 3.4. Gross and Histopathological Analysis

#### 3.4.1. Macroscopic and Radiographic Analysis

Macroscopic evaluation of the samples showed PAG dilation and some intraluminal thrombi. Limited calcified deposits were found on radiography (Figure 5). Radiographic assessments of calcification are shown in Appendix A.

#### 3.4.2. Microscopic Evaluation

The PAG wall appeared thicker than the native pulmonary artery due to the formation of a layer of intimal hyperplasia over the tunica media. The latter one showed a preserved architecture with regularly arranged focally thinned, non-disrupted elastic lamellae without any remarkable differences from the native pulmonary artery tunica media (Figure 6I,J). Smooth muscle cells were regularly distributed among the lamellae with only focal disappearance. Intimal hyperplasia showed several enlarged cells, thin and randomly arranged elastic fibers, and fibrous tissue consisting of variously oriented collagen fibers (Figure 6A–F). The aorta of the SOG always showed a regular arrangement of elastic fibers, as well as the cells distributed among them; no calcifications were found (Figure 6G,H). The presence or absence of necrosis is shown in Appendix A.

#### 3.4.3. Immunohistochemistry

Smooth muscle cells in the tunica media of the PAG were clearly detected, although less represented than the ones in the aortic wall (Figure 7A,B). A continuous vWF-positive endothelial layer was observed on the luminal side. However, intimal hyperplasia cells did not show positivity for this marker (Figure 7C).

A positivity for iNOS was found in the capillaries of the tunica adventitia and in the neocapillaries of intimal hyperplasia (Figure 7E,F). CD45- and CD68-positive inflammatory cells were located near the sutures. Conversely, these markers were almost absent in the PAG’s and aortic walls (Figure 7H,I). The CD45, CD68, and alpha smooth muscle actin (α-SMA)-positive cells both in the PAG’s tunica media and intimal hyperplasia are reported in Appendix A.

#### 3.4.4. Morphometric Analysis

The wall thickness of the PAG was 351.96 ± 171.21 μm, and this was higher than the nearby aorta (113.01 ± 27.39 μm, *p* = 0.003) and the native pulmonary artery (80.30 ± 6.55 μm, *p* < 0.001). The difference was due to the development of intimal hyperplasia (283.13 ± 167.79 μm, *p* < 0.001), while the thickness of the tunica media was not statistically different (68.82 ± 28.25 μm, *p* = 0.31).

Cell density in the PAG was 5028.33 ± 1095.51 cells/mm^2^ in the tunica media and 3925.84 ± 716.99 cells/mm^2^ in intimal hyperplasia (*p* = 0.07). No differences were found between the intimal hyperplasia, the nearby aorta (4930.41 ± 1197.94 cells/mm^2^, *p* = 0.15), and the native pulmonary artery (4255.29 ± 744.15 cells/mm^2^, *p* = 0.38). Similarly, the cell density in the tunica media of the PAG was comparable to the one of the nearby aortae (*p* = 0.85) and the native pulmonary artery (*p* = 0.09).

No differences were found with respect to the aorta (4930.41 ± 1197.94 cells/mm^2^, *p* = 0.19) and the native pulmonary artery (4255.29 ± 744.15 cells/mm^2^, *p* = 0.54).

The density of elastic lamellae in the tunica media of the PAG was 142.05 ± 52.16 lamellae/mm^2^, and no significant difference was found with the nearby aorta (95.45 ± 54.28 lamellae/mm^2^, *p* = 0.11) and the native pulmonary artery (100.16 ± 7.79 lamellae/mm^2^, *p* = 0.1).

Finally, the percentage of fibrosis in the PAG was 52.3% ± 18.0% considering the entire wall, while in the tunica media, it was 29.5% ± 19.6%. Again, no differences were found between the tunica media, the nearby aorta, 23.4% ± 2.0% (*p* = 0.61), and the native pulmonary artery, 29.5% ± 2.3% (*p* = 1).

#### 3.4.5. Transmission Electron Microscopy

The ultrastructural results of the PAG’s tunica media revealed preserved elastic lamellae and collagen bundles. The cells in lamellar units showed a central oval nucleus and an actin filament-rich cytoplasm. In intimal hyperplasia, the elastic and collagen fibers appeared randomly arranged. Smooth muscle cells showed sparse contractile filaments near the cytoplasmic membrane and a large nucleus with moderately dispersed granular chromatin, suggesting poorly differentiated cells (Figure 8).

## 4. Discussion

Among the therapeutic options for congenital aortic stenosis, the Ross procedure represents an interesting strategy, especially in the pediatric population [10]. Indeed, the pulmonary autograft maintains its growth potential, and life-long anticoagulation therapy is not required [11]. However, its main limitation is the potential progressive dilation of the neoaortic root, which increases the risk of aortic valve regurgitation and the consequent incidence of reoperation [12,13]. Several attempts have been made over the years to mitigate this problem, but no approach has been conclusive.

In the current study, we developed a murine model of the Ross operation to investigate the mechanisms involved in the adaptation of the pulmonary vascular root in the systemic position. A PAG was harvested and implanted from syngeneic animals in a heterotopic position (infrarenal abdominal aorta). Similar models involving large animals, such as lambs or sheep, are described in the literature [14,15]. However, a complete evaluation of the histological and mechanical changes in these models has not yet been completely accomplished. Furthermore, large animals limit reproducibility on a large scale due to laborious intra- and post-operative management. Another advantage of a murine model is the possibility of studying the modifications in a medium-long term period by carrying out a shorter follow-up (two months in rats equals about five human years). Different murine models of pulmonary root pressure overload in rats have been also reported, such as pulmonary artery banding [16]. Yet, this kind of pressure overload model would not reflect the effects of a consistent systemic pressure, and the pulmonary root would not undergo separation from the vascular and neurological system (as it happens in a Ross operation).

As previously reported from ultrasound studies performed on patients undergoing Ross surgery, our results showed significant short-term dilation of the graft (+17.7%), followed by mid- to long-term stabilization of the PAG diameter (+3.0% at one month and +7.8% at two months) [17]. It can be hypothesized that part of this dilation may be related to a physiological growth of the PAG, as complete revascularization could be seen on a microscopic evaluation, confirming the preservation of growth potential. Hemodynamic evaluation in the short-term showed lower PSV in the PAG group than in SOG animals, which could be attributed to an increase in PAG diameter causing a deceleration of blood flow. Subsequently, an adaptation of the PAG occurred, which led to loss of compliance, thus mitigating the reduction in blood flow velocity through a structure with an increased diameter. This would explain the comparable PSV in TG and SOG at 2 months.

Our results showed an increase in PAG stiffness. Membrane force values of the PAG were similar to those of the aorta at 1 week and significantly higher at 2 months. The native pulmonary artery always showed the lowest stiffness among the three sample groups. It must be considered that 1 week does not represent sufficient time for any remodeling. The mechanical differences between the PAG and the native pulmonary artery can only be related to immediate dilation after exposure to systemic pressures. This latter phenomenon causes an elongation of the elastic fibers, thus reducing the elasticity of the wall. Therefore, we can assume that the adaptation of the PAG was responsible only for the difference in stiffness observed between 1 week and 2 months. Still, this change is likely not related to an increase in diameter, since ultrasound studies did not show significant dilation after 1 week. Ultrasound and mechanical studies provided equivalent information: PSV and membrane force increased significantly only after the first week of follow-up. As a matter of fact, this period may represent the key phase for any interventions intended to influence the adjustment of the PAG. We can suppose that the increased stiffness of the PAG wall can represent the cause of the progressive dilation of the PAG itself, as also suggested by a study performed in repaired tetralogy of Fallot patients [18].

A histological comparison between the PAG and the native pulmonary artery was performed. This showed development of intimal hyperplasia. This layer is also described in some rare evaluations performed on explanted pulmonary autograft from patients undergoing reoperation after the failure of a Ross procedure [19]. The PAG’s tunica media showed a preserved structure. However, uninterrupted thinning of the elastic lamellae and a focal loss of smooth muscle cells could be observed in some cases. The tunica media fibrosis amount after 2 months was negligible compared to the native pulmonary artery. A continuous endothelium lined the luminal surface, providing a barrier to the migration of inflammatory cells, which were rare.

Intimal hyperplasia caused an increase in wall thickness with a high amount of collagen fibers, reduced and randomly arranged elastic fibers, some capillaries, and many smooth muscle cells. We can assume that these changes represent the PAG response to pressure overload. In fact, despite their common origin, it can be hypothesized that the pulmonary artery and the aorta undergo postnatal modifications, influenced by their environment. Consequently, any changes would activate adaptation mechanisms to cope with the new conditions, which in this case involved the pathological development of subendothelial hyperplasia. In this context, intimal hyperplasia can have important contributions to increased stiffness of the PAG wall described by ultrasound and mechanical studies.

Ultrastructural investigations of the PAG confirmed the preserved tunica media, as well as the disoriented elastic and collagen fibers of intimal hyperplasia and the smooth muscle cells scattered between them. Moreover, a sizeable population of immature cells could also be observed with actin filaments scattered near the cytoplasmic membrane and a large nucleus with dispersed granular chromatin.

The development of intimal hyperplasia is a phenomenon already described in the literature as a response to wall stress and endothelial lesions, such as pulmonary hypertension and venous grafting [20,21,22,23]. In our study, the trigger could probably be the excessive distension of PAG due to pressure overload. The origin of the cells involved in the development of intimal hyperplasia is not yet clear [24]. Our immunohistochemical analysis showed that some cells presented complete differentiation into a smooth muscle cell phenotype, as the cells were intensely α-SMA-positive and showed no cross-reactivity for vWF. Therefore, the cells of intimal hyperplasia may originate from the tunica media and migrate to the intimal layer, where they proliferate before entering a quiescent phase concomitantly with the formation of the superficial endothelial layer [25,26]. However, ultrastructural evaluation of intimal hyperplasia also revealed poorly differentiated cells. In this context, the circulating bone marrow and derived progenitor cells could be a source of cells that contribute to this intimal hyperplasia, as previously reported by other studies [27,28,29,30].

### Limitations

A mixed group of male and female animals was used in our study, and therefore, sex could not be assessed in the analysis.

We decided to not focus our attention on the valve’s leaflets as the main cause of pulmonary autograft failure since the Ross operation is related to vessel walls.

A possible difference in blood flow streaming may exist between the ascending and the abdominal aorta, but its consequences on the PAG remodeling could not be assessed. However, the pressure stress within the arterial blood tree remained the same as well as the pressure stress on the arterial wall, and we believe that the main determining factor for the PAG remodeling was the difference in pressure between the pulmonary and the systemic circulation.

Another limitation of the present study is the lack of a short-term histopathological evaluation that would have allowed correlation with ultrasound and mechanical studies. In this way, a more accurate characterization of the origin of the cells of intimal hyperplasia could have been performed. Further studies with shorter follow-up times may likely answer questions related to this. Additionally, other cellular markers may be needed to better characterize the immature smooth muscle cells of the neointima in order to identify the type of their precursor. Moreover, calcification related-protein markers may help to investigate the mineralization process in the tissue.

## 5. Conclusions

In conclusion, we were able to establish a reliable a murine model of the Ross operation which is reproducible on a large scale and allows the assessment of the PAG’s response to pressure overload. A rapid dilation of the PAG was observed after the implantation with consequent medium- and long-term structural changes, resulting in an increase in the stiffness of its vascular wall. Histopathological studies showed the development of intimal hyperplasia and preserved but thinned elastic lamellae in the tunica media, which can possibly justify the increased stiffness.

We can speculate that the development of intimal hyperplasia can represent a maladaptive response of the PAG to pressure overload that can lead to neoaortic valve regurgitation observed in patients after the Ross procedure, possibly because of a physical impediment and a structural mechanism. As for the first cause, it is possible that its thickness can affect the valve leaflets’ motion, while the reduced compliance caused by intimal hyperplasia can decrease the elastic recoil of the vascular wall and predispose to progressive dilation under the stress of the systemic pressure, causing leaflet coaptation failure. The correlation with non-invasive ultrasonographic studies and the maladaptive structural PAG adaptation can be useful in monitoring the post-operative adaptation of the PAG and the timing of its failure. Further studies aimed at inhibiting this process are needed to define the role of these findings and possibly improve the outcome of the Ross operation in the pediatric population.

## Figures and Tables

**Figure 1 jcm-11-03742-f001:**
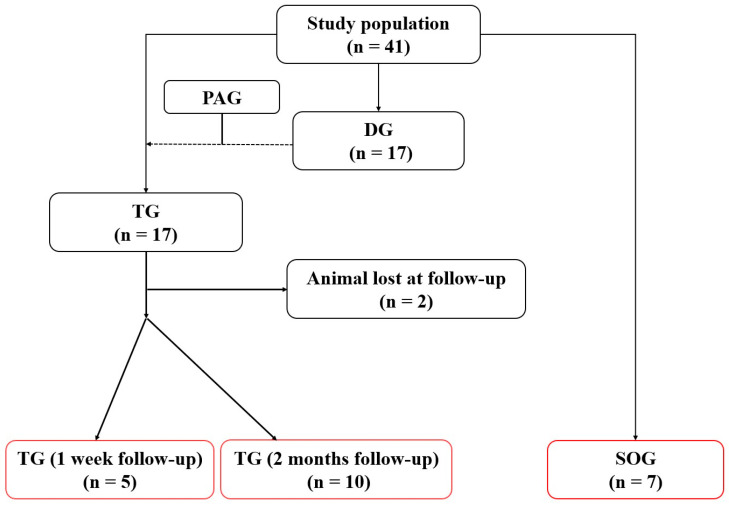
Flowchart of the study, showing the three groups and their numerosity. DG: donor group; PAG: pulmonary artery graft; SOG: sham-operated group; TG: transplant group.

**Figure 2 jcm-11-03742-f002:**
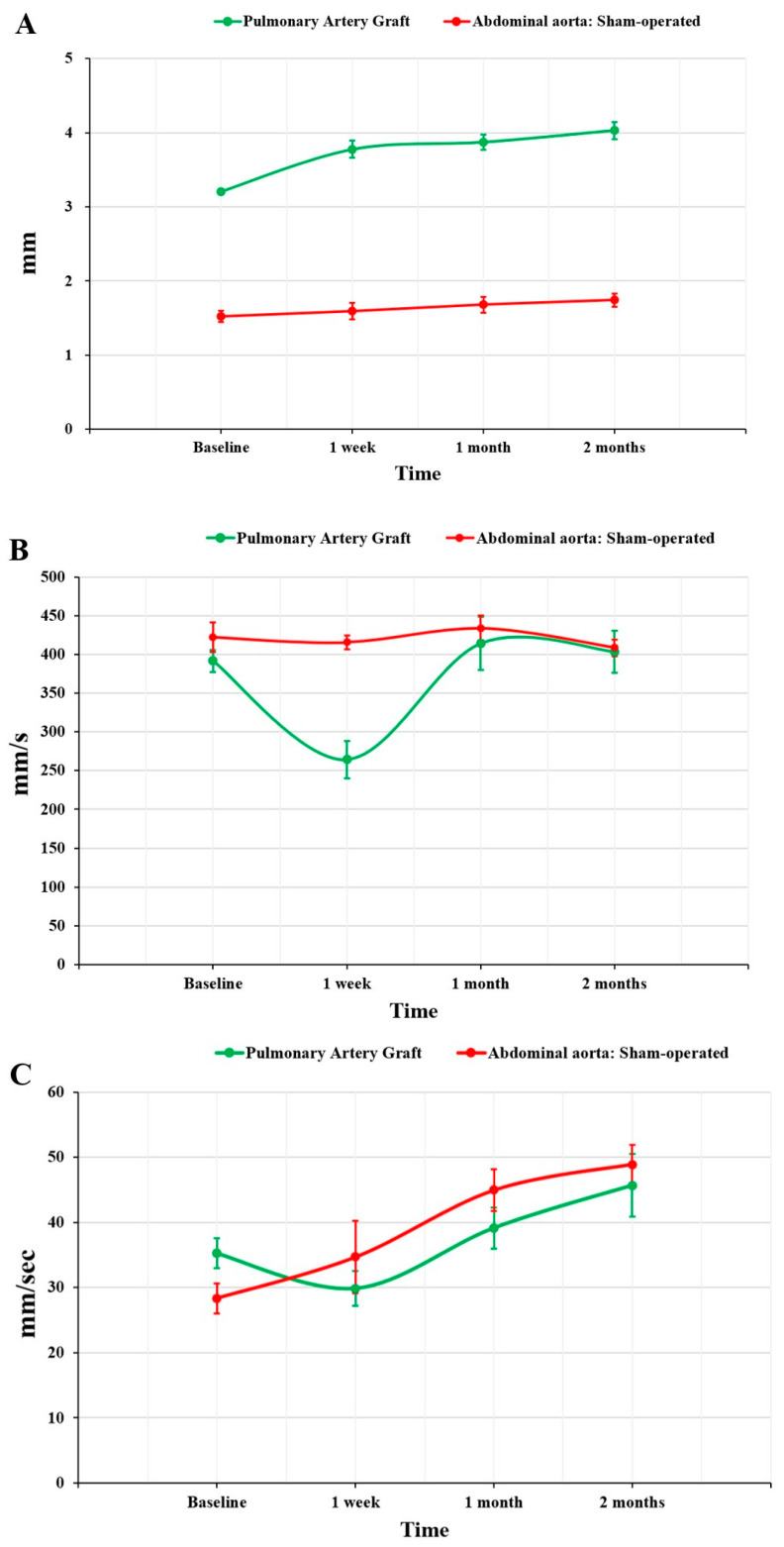
Graphs showing the comparison of the ultrasound parameters between the two groups over the study period. (**A**) Diameter; (**B**) peak systolic velocity; and (**C**) end-diastolic velocity.

**Figure 3 jcm-11-03742-f003:**
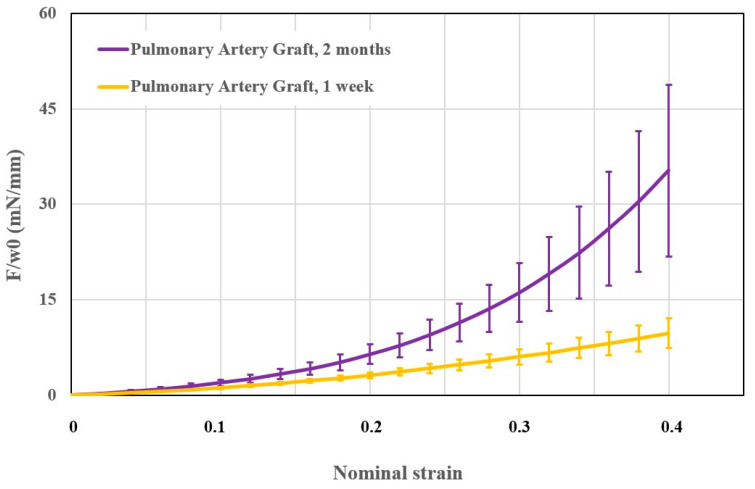
Graph showing the comparison of PAG mechanical properties at 1 week (yellow, n = 5) and 2 months (purple, n = 10). The x-axis shows the nominal strain and the y-axis the membrane force. Data presented as mean ± standard error of mean. F: tensile force; w0: initial width.

**Figure 4 jcm-11-03742-f004:**
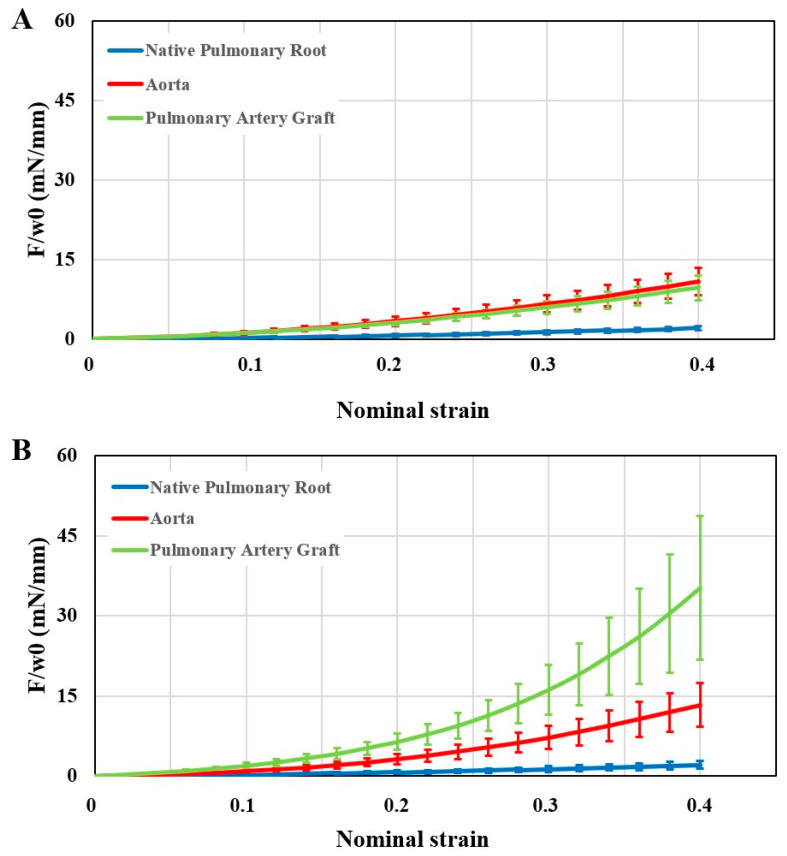
Graphs showing stiffness of the aorta (red), native pulmonary artery (blue), and PAG (green) at 1 week ((**A**), n = 5) and 2 months ((**B**), n = 10) after surgery. The x-axis shows the nominal strain and the y-axis the membrane force. Values are expressed as mean ± standard error of the mean. F: tensile force; w0: initial width.

**Figure 5 jcm-11-03742-f005:**
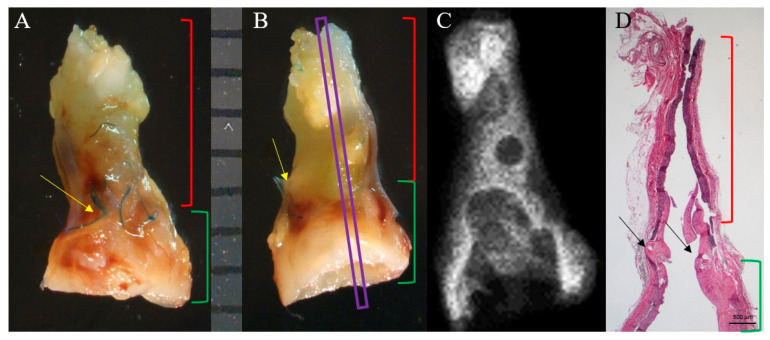
(**A**,**B**) Macroscopic images of the PAG and the near aorta 69 days after surgery. The PAG (green stapes) appeared more dilated than the infrarenal abdominal aorta (red stapes). Yellow and black arrows indicate the sutures. (**C**) Radiography of the explant: no calcifications were visible. (**D**) Histological image of the entire explant in correspondence to the purple section in (**B**); the same portions indicated in (**A**,**B**) are highlighted. Weigert-Van Gieson, original magnification 12.5×.

**Figure 6 jcm-11-03742-f006:**
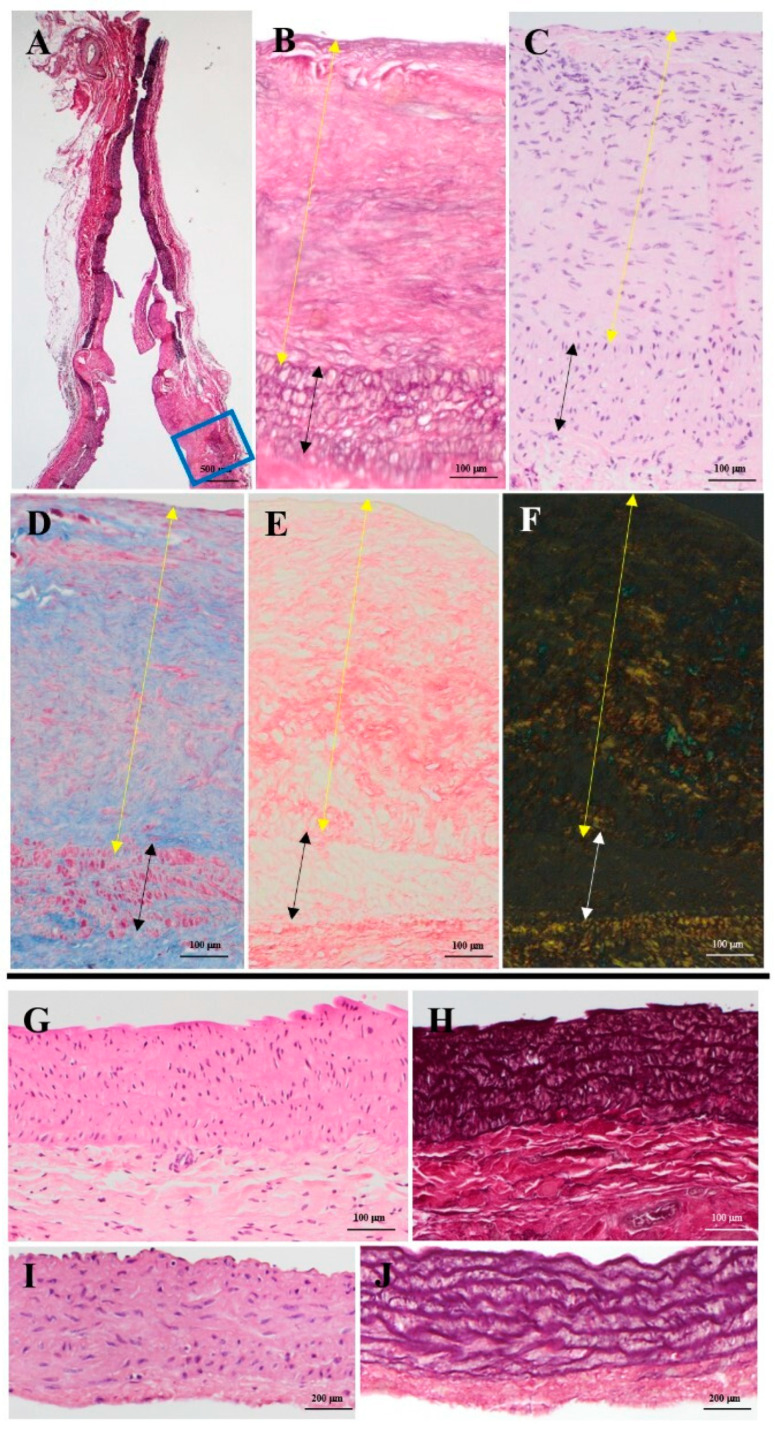
(**A**) Same section shown in Figure 5D. (**B**–**E**) Detail of the PAG wall highlighted by the blue box in (**A**). The black and white arrows indicate the tunica media, while the yellow ones indicate intimal hyperplasia. Native tunica media was visible and showed preserved although thinned elastic fibers (**B**), conserved cell density (**C**), scarce fibrosis (**D**), and regularly oriented collagen fibers (**E**,**F**). Intimal hyperplasia showed disorganized elastic fibers (**B**), hypercellularity (**C**,**D**), and collagen fibers (**D**) arranged in various directions of space (**E**,**F**). (**G**,**H**) SOG aorta; no calcifications were observed. (**I**,**J**) Native pulmonary artery; note the regular disposition of elastic fibers and the cells among them. (**A**,**B**) Weigert-Van Gieson, original magnification 12.5× (**A**), 100× (**B**); (**C**) hematoxylin-eosin, original magnification 100×. (**D**) Azan-Mallory, modified Heidenheim, original magnification 100×. (**E**,**F**) Picrosirius red in transmitted light, (**E**) in polarized light, (**F**) original magnification 100×. (**G**) Hematoxylin-eosin, original magnification 100×; (**H**) Weigert-Van Gieson, original magnification 100×; (**I**) hematoxylin-eosin, original magnification 125×; (**J**) Weigert-Van Gieson, original magnification 125×.

**Figure 7 jcm-11-03742-f007:**
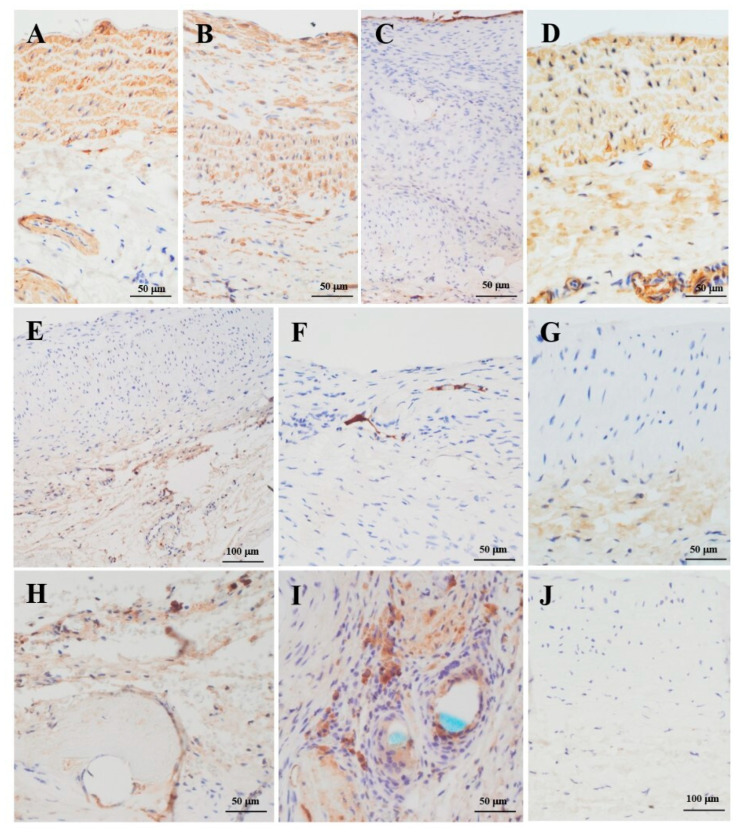
Detail of the abdominal aorta (**A**) and the graft wall (**B**,**C**,**E**,**F**,**H**,**I**) of the same section shown in Figure 5D, and SOG aorta (**D**,**G**,**J**). (**B**) In the tunica media of the PAG, the 70% of cells were α-SMA-positive. Many large α-SMA-positive cells, 90% overall, can be seen in intimal hyperplasia. (**C**) The luminal surface of the PAG shows a continuous endothelial layer. (**E**,**F**) Capillary endothelial cells in the adventitia were positive for the iNOS marker (**E**), as were some neocapillaries of intimal hyperplasia (**F**). (**H**,**I**) Inflammatory cells were found predominantly near the sutures; CD45-positive cells accounted for about 1% of all cells (**H**), and CD68-positive cells for 5% (**I**). (**D**,**G**,**J**) SOG aorta tunica media cells were 98% α-SMA-positive (**D**); some iNOS-positive (**G**) and a negligible amount of CD45- and CD68-positive cells (**J**) were detected. (**A**,**B**,**D**) anti-α-SMA, original magnification 200×. (**C**) anti-vWF, original magnification 200×. (**E**–**G**) anti-iNOS, original magnification 100× (**E**); 200× (**F**,**G**). (**H**) anti-CD45, original magnification 200×. (**I**,**J**) anti-CD68, original magnification 200× (**I**); 100× (**J**).

**Figure 8 jcm-11-03742-f008:**
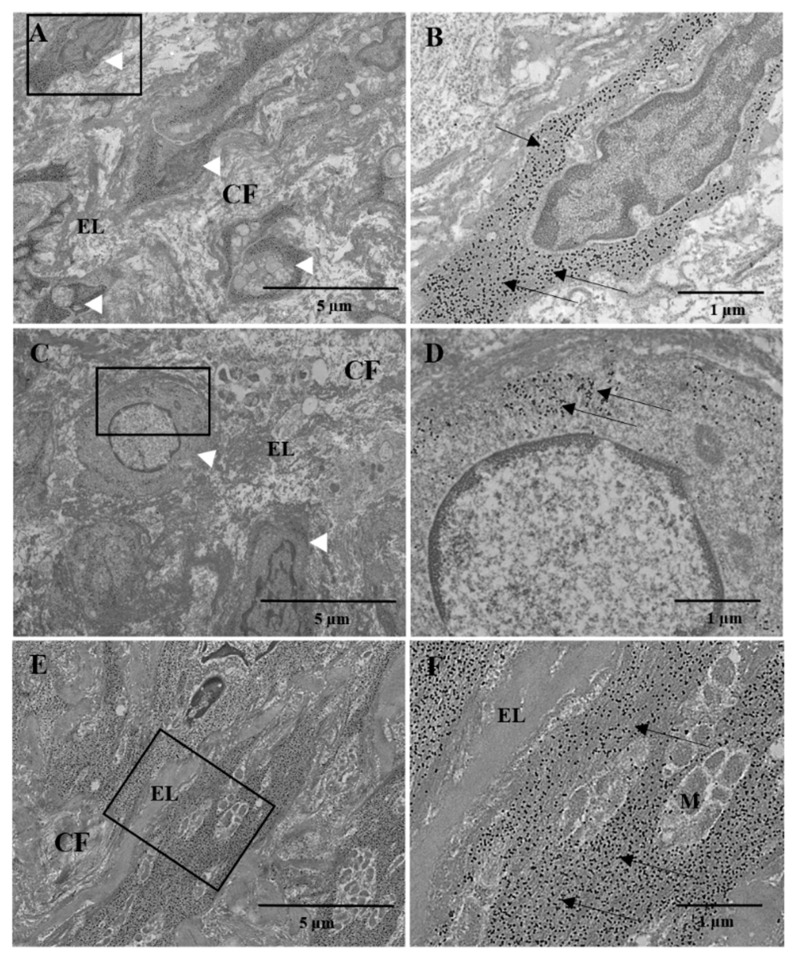
(**A**) Tunica media of the PAG 69 days after implantation. Smooth muscle cells of regular and oriented shape (arrowheads) were interspersed with elastic fibers and collagen; (**B**) close-up of (**A**) (black box) in which the cytoplasmic α-actin filaments of a smooth muscle cell were detected by 10 nm colloidal gold particles (arrows). (**C**) Intimal hyperplasia of PAG: elastic and collagen fibers appeared disoriented, and many smooth muscle cells (arrowheads) showed an immature appearance; (**D**) close-up of (**C**) (black box). Smooth muscle cell with a large nucleus and few contractile filaments close to the cytoplasmic membrane (arrows). (**E**) SOG aorta tunica media. Smooth muscle cells appeared regularly orientated and commixed with elastic fibers and collagen bundles. (**F**) Close-up of (**E**) (black box). Smooth muscle cells showed typical α-actin filaments in cytoplasm (arrows). CF: collagen fibers; EL: elastic fibers; M: mitochondria. (**A**–**F**) Post-embedding immunogold α-SMA labeling. (**A**,**C**,**E**) 3000×, original magnification; (**B**,**D**,**F**) 10,000×, original magnification.

## Data Availability

The data presented in this study are available on request from the corresponding author.

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
