# Peer review of "Mechanical and Structural Adaptation of the Pulmonary Root after Ross Operation in a Murine Model"

_jcm, 2022, doi:10.3390/jcm11133742_

Round 1

Reviewer 1 Report

The author reported their new murine model for pulmonary root adaptation after being implanted in the abdominal aorta, mimicking post-Ross operation pressure conditions.

The author designed a thoughtful and detailed method for this model, which makes it easy to be reproduced and advanced. 

The analysis of the result moved from invivo condition and mechanical tissue evaluation all the way to the cell level analysis.

The work in developing the model and analysis is very interesting and can add a significant contribution to answering the addressed question and expanding it to other applications.

I have a few questions and comments; I hope they can help and benefit the manuscript.

1.     Under pulmonary harvesting, what anesthetic protocol (drugs, ventilation...etc.) did you use? The method needs to be clear in the text to comply with animal care.

2.     In line 73: Which method do you use to confirm the heart stop beating, EKG or visual confirmation? Also, you need to state that you are using cooling as your euthanasia method for this group unless you used a different method.

3.     In line 78: what refrigerator temperature do you use to store the pulmonary graft?

4.     Under receipt operation, line 83, how did you maintain the anesthesia during the procedure with spontaneous breathing? Did you give inhalation through an endotracheal tube or mask, or did you give injections?

5.     Similar to the harvesting group, the Sham-operated group has no anesthesia details; you can either add a separate section for animal anesthesia that you used in all groups if it was similar or add the details under each group.

6.     Under postoperative management, I believe you used the same tissue preservation process for the sham group, too, correct? (Not clear in the methods)

7.     In lines 173-174, you used room temperature at 25 °C for the testing condition instead of 36-37 °C, which is a body temperature in murine. How do you think that can affect your testing results? 

8.     In lines 187-189, the mentioning non-published data to justify your cut-off point is unnecessary or relevant to the paper. You can mention you elected this cut-off safety value based on your observation. Otherwise, you need to provide the non-published data in this manuscript.

9.     In ex-vivo mechanical testing, why did you choose to analyze the strain against the forces? While you have the material thickness, why not analyze the stress-strain relation, which is more informative and can add higher value to your paper? It would be great if you could add it to this manuscript.

10.  Did you look at the graft valve leaflets? Is there any comment you can add about that? Also, I think you need to add the limitations that the model cannot be used to evaluate the pulmonary graft valve.

11. Another limitation is the difference in blood flow streaming between the ascending and abdominal aorta, which may show a non-uniform effect and change in the ascending aorta position compared to the abdominal.

           Thank you

Reviewer 2 Report

The manuscript by Claudia Cattapan et al. is aimed to examine the mechanical and structural adaptation of the pulmonary root wall under increased pressure load. However, the study was not well designed for it’s an observational study without any molecular mechanism.Mostly they only provided representative images without showing the quantification. Also the control (sham) group  was missing in several experiments. Please find my comments in details as below that need to be addressed before further consideration by the journal.

  1. More details on Root operation should be provided in the introduction section. 
  2. Why did the author perform root operation on Lewis rats instead of other strains? Is there any specific reason for the priority of this strain?
  3. Why were rats in TG group all male and those in DG group all female? Is there any rationale for the design?
  4. Since the authors detected an increase in PAG diameter, I was wondering whether they examine the pulmonary hemodynamics (for example, systolic pulmonary arterial pressure) with echocardiography at baseline and during the follow-up?
  5. It’s uncommen to lable the panel (A, B, C) in Figure 1 in the lower left position. It’s better to put it in the upper left cornor.
  6. What’s the reason that peak systolic velocity was declined at 1 week follow-up in rats with pulmonary artery graft and then difference was abrogated at 2 weeks post-surgery.
  7. In Figure 3, why didn’t they test the mechanical response of the aorta in the sham operation group? How many rats were tested for ex vivo mechanical tests? Please speficy the rat numbers in each condition. There was no description whether there was any significant differences between the groups. If so, please add the P value. 
  8. In Figure 4, please adjust the y-axis (eg. with a range of 0 – 30 mN/mm) for better visualization of the difference. Again, please speficy the rat numbers in each condition in Figure legends.
  9. It would be better to add scale bar to the histological images.
  10. A comparison of the representative and quantification of SMA-postive cells, CD45+, iNOS+ and CD68+ between PAG group (1 week and 2 month after surgery) and control group should be added into the main Figures.
  11. The control group in Figure 8 was also missing.
  12. Did they detect any calfication related protein (eg. BMP2/4/6 proteins, OPN, OPG or other markers) changes between PAG group and sham group? 

Reviewer 3 Report

What are the causes for the initial drop in PSV and progressive continued rise in PSV thereafter? Initial dilation, progressive dilation, reduced compliance vs increased compliance, and / or elasticity? These could be assessed to increase the impact of these data. 

Other factors affecting PSV are cardiac output, heart rate, etc. Were these evaluated to assess how they change post graft implantation to see what adaptive / maladaptive changes occur. PSVR may be a better predictor in an attempt to control for CO / heart rate / rhythm factors. 

PSV remains relatively constant in the SOG group. Why does PDV increase in both groups?

Abdominal blood flow is more laminar than flow in the ascending aorta. As a result, different changes may occur in a homograft in the descending versus ascending aortic positions. 

Were any surgical related stenoses at the anastomosis site noted? 

PA grafts were larger in diameter than the abdominal aorta. This affects wall tension and stress, which is much higher for PA grafts versus the abdominal aorta. 

 Line 512, please speak on how reduced elasticity likely owing to the small medial layer of the PA graft contribute to increase stiffness, thereby leading to progressive dilation (Seki, Mitsuru, et al. "Progressive aortic dilation and aortic stiffness in children with repaired tetralogy of Fallot." Heart and vessels 29.1 (2014): 83-87). There is literature data to suggest that this may be the culprit for progressive dilation.

Round 2

Reviewer 2 Report

The manuscript by Claudia Cattapan et al. is aimed to examine the mechanical and structural adaptation of the pulmonary root wall under increased pressure load. I congratulate the authors for making the necessary revisions or providing explanations and clarifications to my previous concerns. Therefore, I have no further questions at present and would like to recommend accepting the current version.